# Understanding Aesthetics with Language: A Photo Critique Dataset for Aesthetic Assessment

**Daniel Vera Nieto**
Media Technology Center (MTC)
ETH Zurich
Zurich, CH
daniel.veranieto@inf.ethz.ch

**Luigi Celona**
DISCo
University of Milano-Bicocca
Milano, IT
luigi.celona@unimib.it

**Clara Fernandez-Labrador**
Media Technology Center (MTC)
ETH Zurich
Zurich, CH
clabrador@inf.ethz.ch

## Abstract

Computational inference of aesthetics is an ill-defined task due to its subjective nature. Many datasets have been proposed to tackle the problem by providing pairs of images and aesthetic scores based on human ratings. However, humans are better at expressing their opinion, taste, and emotions by means of language rather than summarizing them in a single number. In fact, photo critiques provide much richer information as they reveal how and why users rate the aesthetics of visual stimuli. In this regard, we propose the Reddit Photo Critique Dataset (RPCD), which contains tuples of image and photo critiques. RPCD consists of 74K images and 220K comments and is collected from a Reddit community used by hobbyists and professional photographers to improve their photography skills by leveraging constructive community feedback. The proposed dataset differs from previous aesthetics datasets mainly in three aspects, namely (i) the large scale of the dataset and the extension of the comments criticizing different aspects of the image, (ii) it contains mostly UltraHD images, and (iii) it can easily be extended to new data as it is collected through an automatic pipeline. To the best of our knowledge, in this work, we propose the first attempt to estimate the aesthetic quality of visual stimuli from the critiques. To this end, we exploit the polarity of the sentiment of criticism as an indicator of aesthetic judgment. We demonstrate how sentiment polarity correlates positively with the aesthetic judgment available for two aesthetic assessment benchmarks. Finally, we experiment with several models by using the sentiment scores as a target for ranking images. Dataset and baselines are available[1].

## 1 Introduction

Automated Image Aesthetic Assessment (IAA) is a widely discussed topic in the computer vision community and is receiving an increasing attention due to the explosive growth of digital photography. The literature reports mainly on predicting aesthetic preference in close agreement with human judgement. In particular, most studies deal with IAA in terms of high vs. low aesthetic quality [25],

---

[1]https://github.com/mediatechnologycenter/aestheval

36th Conference on Neural Information Processing Systems (NeurIPS 2022) Track on Datasets and Benchmarks.

regression of the aesthetic score [14], and prediction of the distribution of the aesthetic ratings [37]. Various datasets were collected to contribute developing and evaluating the previous studies. These datasets consist of images annotated with aesthetic scores. However, summarizing the aesthetic judgment in a single value limits the representation of visual aesthetics. First, aesthetic scores are highly dependent on the voting procedure (i.e., voting scale, number of stimuli, questions and adjectives in the voting scale). Second, it has been shown that they might provide a variable or even negative impact on the prediction of human behavior and thus on the success of social content [33]. Third, aesthetic scores do not provide any interpretability of why an image is aesthetically pleasing or not. Thus, it makes sense to annotate images with richer high-level aesthetic attributes [20] or aesthetic criticism captions. Many image-sharing sites, e.g., Flickr, Photo.net, and Instagram, support user comments on images, allowing rating explanations. User comments usually introduce rationale about how and why users evaluate the aesthetics of an image. Comments such as "good composition", "vivid colors", or "a fine pose" are more informative than ratings for expressing pleasing photographic aspects. Similarly, comments such as "too dark" and "blurry" explain why low ratings occur.

On the basis of previous considerations, our first contribution is the Reddit Photo Critique Dataset **RPCD**, a collection of high resolution images associated with photo critiques (i.e., 74K images and 220K comments). The dataset has been obtained from a Reddit community[2] of photography amateurs whose purpose is to provide feedback to help amateurs and professional photographers improve. Figure 1 shows some samples from our RPCD. The dataset presented in this work differs in many ways from existing photo critic datasets. First, the images are mostly FullHD as they were captured with recent photo sensors and imaging systems. Secondly, the proposed dataset is among the largest in terms of the number of images-comments. Third, the comments of our dataset are on average longer and more informative (basing on the score proposed in [10]) than those of the previous datasets.

In the literature, IAA models are trained on datasets in which each image is associated with a rating. However, the problem of how to obtain a rating for IAA without requiring human intervention given a dataset with image-comment pairs is not addressed. The degree of emotion and valence of critique comments is an excellent indicator of the success of contents on social media [33]. Therefore, together with the dataset, we present a new solution to rank images by exploiting the polarity of criticism as an indicator of aesthetic judgments. To the best of our knowledge, this is the first attempt to leverage image critiques to define a score for IAA.

Finally, we design a framework to evaluate different methods on the proposed dataset and other aesthetic critique datasets in the literature on the image aesthetic assessment and aesthetic image captioning tasks.

We find that: (i) the aesthetic scores and the proposed sentiment scores are positively correlated on two photo critique datasets annotated with both comments and scores; (ii) Vision Transformer (ViT) surpasses state-of-the-art methods for image aesthetic assessment; (iii) learning aesthetics-aware features produces a significant increase in performance over using semantic features. This behavior also occurs for models with the same architecture but trained for different purposes.

## 2   Related Work

In this section we briefly analyze the main datasets and methods for the two tasks of image aesthetic assessment and aesthetic critique captioning. We refer the reader to [42] for a comprehensive review on computational image aesthetics.

**Image Aesthetic Assessment.**   For the design and evaluation of Image Aesthetic Assessment (IAA) methods, the construction of the aesthetic image evaluation benchmark dataset has become the fundamental prerequisite for the research. Many datasets were collected in which subjective aesthetic quality scores were acquired for each image. The acquisition of subjective scores can be realized through manually scoring experiments in the lab [25], online scoring on image sharing website [20, 28], and crowdsourcing evaluation [36]. Methods that exploit the previous datasets for aesthetic assessment can be divided into model-based [7, 27, 43] and data-driven [4, 14, 24, 37]. While model-based methods rely on hand-crafted features to model aspects such as the Rule of Thirds,

---

[2]`www.reddit.com`

depth of field, colour harmony, etc., the data-driven methods usually train CNNs on large-scale datasets to predict an overall aesthetic rating.

**Aesthetic Critique Captioning.** The first work on Aesthetic Critique Captioning, also known as Aesthetic Image Captioning (AIC), presents the so called Photo Critique Captioning Dataset (PCCD) based on a professional photo critique website[3] and a method for predicting aspect-centric captions [5]. The other AIC datasets in the literature are obtained by crawling images together with their comments from an on-line community of photography amateurs[4]. AVA-Comments [45] extends AVA to include all user comments for images, while AVA-Captions [10] filters original AVA photo comments to keep only the most useful. Finally, DPC-Captions [17] contains 154,384 images and 2,427,483 comments. Each comment is automatically annotated with one of the 5 aesthetic attributes of the PCCD through aesthetic knowledge transfer. Few AIC methods are present in the literature for predicting aesthetic comments [10, 41], aspect-centric aesthetic captions [5], or simultaneously the aesthetic rate and an aesthetic caption [38].

## 3 Background and Theory

In this section we provide a formal definition for the classical image aesthetic assessment problem and describe our novel formulation of the problem.

**Notations.** We represent sets and matrices with special Latin characters (e.g., $\mathcal{M}$) or bold Latin characters (e.g., M). Lower or uppercase normal fonts, e.g., $K$ denote scalars. Lowercase bold Latin letters represent vectors as in v. We use lowercase Latin letters to represent indices (e.g., $i$).

### 3.1 Image Aesthetic Assessment

Image Aesthetic Assessment (IAA) methods aim at computationally judging the aesthetic value of images based on human ratings and photographic principles. These methods map an input image $I_i \in \mathbb{R}^{H \times W \times 3}$ to an aesthetic score $s_i$ and can be divided into binary classification methods which predict a single binary score $s \in \{0, 1\}$, and regression methods which predict a single real score $s \in \mathbb{R}$ or a probability distribution of scores $p(s)$. Classification methods are used to distinguish "good" from "bad" images whereas regression methods are preferred to rank collections of images. These methods typically rely on public datasets [20, 25, 28] that contain $N$ pairs of images and aesthetic scores such that $\mathcal{D} = \{(I_1, s_1), \ldots, (I_N, s_N)\}$, where the ground truth score per image is computed as the average rating given by $K$ human raters:

$$s_i = \frac{1}{K} \sum_{k=0}^{K} s_k, \tag{1}$$

The scores are further thresholded by the mid-point of the rating scale for the classification task. However, asking people to evaluate the aesthetic value of an image with a single global score is very challenging and can be extremely biased by the content of the images. Additionally, these scores alone do not provide any explicit information about the reasons behind the voting.

### 3.2 Aesthetic Critiques

Recent datasets [5, 10, 17] extend the IAA problem including captions related to photo aesthetics and/or photography skills. These datasets contain $N$ images each described by $K$ aesthetic critiques c such that $\mathcal{D} = \{(I_1, c_1^1, \ldots, c_1^K) \ldots, (I_N, c_N^1, \ldots, c_N^K)\}$. Common critiqued aesthetic aspects are composition, subject of photo, use of camera or color. In this context, novel algorithms have been developed to generate aesthetic-oriented critiques for images. Therefore, these methods map an input image $I_i \in \mathbb{R}^{H \times W \times 3}$ to an aesthetic critique $c_k$. While photo critiques give explicit feedback about why images are aesthetically pleasing or not, it has not been explored yet how to exploit such critiques for classification or image ranking, which is the ultimate goal of IAA. Furthermore, critique

---

[3]https://gurushots.com/
[4]https://www.dpchallenge.com/

generative models present an additional challenge with respect to conventional captioning models due to the subjective nature of problem at hand. For this reason, many generated critiques tend to express critic's preference (e.g., "I like the colors" and "nice photo") rather than providing a detailed opinion or critique of the image aesthetics.

## 3.3 Leveraging Aesthetic Critiques for Image Aesthetic Assessment

Ranking images from aesthetic critiques comes as a natural extension of the problem. In this paper, we are interested in leveraging the interpretability given by image captions to automatically discover the aesthetic score that truly defines the images beauty. Given an input image $I_i \in \mathbb{R}^{H \times W \times 3}$ and $K$ aesthetic critiques associated to the image, we propose to use sentiment polarity analysis on each critique $c_k$ to define the aesthetic score $s_i$ of the image. Sentiment polarity for a comment defines the orientation of the expressed sentiment, i.e., it determines if the text expresses the negative, neutral or positive sentiment of the user about the entity in consideration. A sentiment polarity model maps a given critique $c_k$ to a vector $p \in \mathbb{R}^3$ which can be interpreted as the probabilities of the given critique to express a negative, neutral or positive feeling with respect to the aesthetic value of the image. We define the sentiment score $s_k$ of a critique as follows:

$$s_k = \frac{\sum_{l=0}^{2} p_l l}{2},\tag{2}$$

Where $l$ is the label for negative, neutral or positive sentiment respectively and $p_l$ the probability associated to the label. The sentiment scores for all the critiques of an image are then averaged to obtain an overall sentiment score $s_i$ as in Eq. 1. The proposed dataset can then be defined such that $\mathcal{D} = \{(I_1, c_1^1, \ldots, c_1^K, s_1) \ldots, (I_N, c_N^1, \ldots, c_N^K, s_N)\}$.

To the best of our knowledge, this is the first attempt to estimate the aesthetic quality of visuals directly from critiques rather than from human ratings. Our proposal is significant for several reasons. Critiques are an important indicator of human judgment, generally more valuable than simple ratings as they provide an explanation of why a visual is aesthetically pleasing or not [33]. However, critiques are unstructured data that do not directly indicate the level of aesthetic appreciation. Therefore, our proposed score is a way to obtain a compact and quantifiable representation of the level of appreciation of an image inferred from the critiques. Second, thanks to our proposal, two related aesthetic tasks are linked. Indeed, the datasets created for Aesthetic Image Captioning (AIC) can be applied to the design of models for both AIC and Image Aesthetic Assessment (IAA). The integration of the two tasks is useful because the critiques guarantee the explainability of a score; on the other hand, the ratings might allow the prediction of valence-sensitive critiques. Finally, our proposal consists of a weakly-supervised labeling approach which has the advantage of requiring human intervention solely to provide comments on the image. Existing datasets such as PCCD and AVA demanded intensive human effort to provide ratings and comments.

# 4 RPCD: Reddit Photo Critique Dataset

The Reddit Photo Critique Dataset (RPCD) is a collection of high-resolution images associated with photo critiques obtained by the Reddit communities. We first give a description of the dataset collection and statistics in Section 4.1. Section 4.2 details how we automatically rank the images following the criticism-based approach described in Section 3.3. Finally, in Section 4.3, we thoroughly analyze the images and comments present in our dataset.

## 4.1 Dataset Collection and Statistics

**Collection Modality.** For the collection of the RPCD dataset, we identified Reddit communities used by amateur and professional photographers to upload their images or to discuss about photography. In particular, the following six communities (known as subreddits) were identified: /r/AskPhotography, /r/photocritique, /r/photographs, /r/portraits, /r/postprocessing, /r/shittyHDR. After a careful review of the different subreddits, we selected the /r/photocritique subreddit with a total of 168,222 posts and 731,772 comments. The

decision was made based on the rules of the community[5] which makes its content specially suitable for the task at hand. Namely, it mostly contains posts with amateur and professional images that get feedback from other photographers and hobbyists. We downloaded all the posts and comments from the selected subreddit between May 2009 and February 2022 using the Python Reddit API Wrapper (PRAW[6]) and the Pushshift platform [2]. Nevertheless, we still note that the other subreddits may hold relevant information that can also be exploited in further works. See Appendix A.1 for more details regarding the number of posts/comments per year in the aforementioned subreddits.

**Automatic Filtering.** The selected posts are then filtered by using an automated pipeline designed to be reused over time or for other communities. It consists of the following steps. First, each post consists of an image along with a description provided by the photographer usually explaining the aesthetic intent of the photograph and the technical details of the camera used. Additionally, each post has comments from other users structured as layered conversations. As required by the subreddit rules, the first level comments must be a critique to the image in the post. Therefore, we keep the top level comments since they are actual critiques and they are not a follow up comment or part of the body of a conversation. The description and the comments under the first level are discarded, thus reducing the number of comments from 731,772 to 284,426. Secondly, we remove the posts with no comments or whose image is no longer available. As a result, the number of posts is reduced to 103,190. Finally, filtering posts with corrupted or placeholder images leads to the final dataset consisting of 73,965 posts, each of them consisting of an image and an average of 3 critiques to that image.

**Statistics.** Our RPCD dataset consists of 73,965 images with a resolution of $2993 \times 2716$ pixels on average. A total of 219,790 photo critique comments is available, with an average of 49.1 words per comment, a standard deviation of 55.5 and a maximum of 1286 words. Each image has an average of 2.9 comments associated with it, with a standard deviation of 3.7. The general information of our dataset and a comparison with related datasets is presented in Table 1. Several considerations can be made. First, our RPCD dataset is the first large-scale photo critique dataset, with a $\sim 17$ times and $\sim 7$ times increase in the number of images and comments, respectively, compared to the previously available photo critique dataset, PCCD [17]. Secondly, it has a slightly higher average length of comments with respect to PCCD and about 3 times that of AVA-Comments [45] (hereafter simply referred to as AVA). This increase in the amount of information opens the door to the use of large language models to exploit the unstructured information available in form of text. Third, our RPCD dataset consists of images with a much higher resolution than previous datasets (especially those obtained from `DPChallenge.com`). See Appendix A.2 for a detailed comparison. This may be due to the difference in the time periods of collection for the different datasets. For example, AVA dataset has images posted only until 2011, when the technical performance and availability of cameras were inferior to nowadays. Consequently, the aesthetic quality is very likely to depend on the perceived technical quality [18].

## 4.2 Sentiment polarity prediction

We propose to use sentiment polarity analysis on the aesthetic critiques to define the aesthetic scores, a.k.a sentiment scores, of the images as detailed in Section 3.3. Sentiment analysis methods can be categorized into lexicon-based methods [16, 35], machine learning methods [11, 44], and hybrid methods [29]. Recently, deep learning methods have enabled the design of sentiment analysis models that have achieved impressive performance over the previous methods [3, 23, 31, 34].

In this work, we use TwitterRoBERTa [1], a deep learning based method inspired by RoBERTa [23], to extract the sentiment polarity on aesthetic critiques. TwitterRoBERTa achieved the best trade-off between performance and model complexity among all the models that participated in the Sentiment Analysis in the Twitter challenge [34]. Although the model is trained in a different domain (Twitter), we assume that the domain is similar enough (social media) to use this model. Future work could explore the use of models tailored for the Reddit sub-domain. Additionally, Transformer models fine-tuned for sentiment analysis are known to have their own set of bias [7]. Hence, a deeper analysis

---

[5] `https://www.reddit.com/r/photocritique/`

[6] `https://github.com/praw-dev/praw` (Accessed on 06/05/2022).

[7] `https://huggingface.co/distilbert-base-uncased-finetuned-sst-2-english#risks-limitations-and-biases`

Table 1: Comparison of the properties in different benchmark datasets on image aesthetic captioning.

| Dataset | AVA-Comments [45] | DPC-Captions** [17] | PCCD [5] | RPCD (Our) |
|---|---|---|---|---|
| Images | 253,961 | 117,132 | 4,235 | 73,965 |
| Avg image resolution | 607×537 | 606×534 | 1414×1202 | 2993×2716 |
| Attributes | – | 5 | 7 | 7* |
| Comments | 3,601,761 | 208,926 | 29,645 | 219,790 |
| Comments per image | 14.1 | 1.8 | 6.6 | 2.9 |
| Avg words per comment | 14.6 | 24.5 | 41.1 | 49.1 |
| Max words per comment | 2146 | 549 | 780 | 1286 |
| Content category | 66 | 66 | 27 | 6* |
| Rating scale | 1-10 | 1-10 | 1-10 | 0-1* |
| Avg raters per image | 6 | 15 | 7 | – |

*The aspect is obtained through machined-based annotation. See Appendix A.4.
**The figures reported on this table are produced using the code made available by the authors of the dataset and differ from those stated in the original paper.

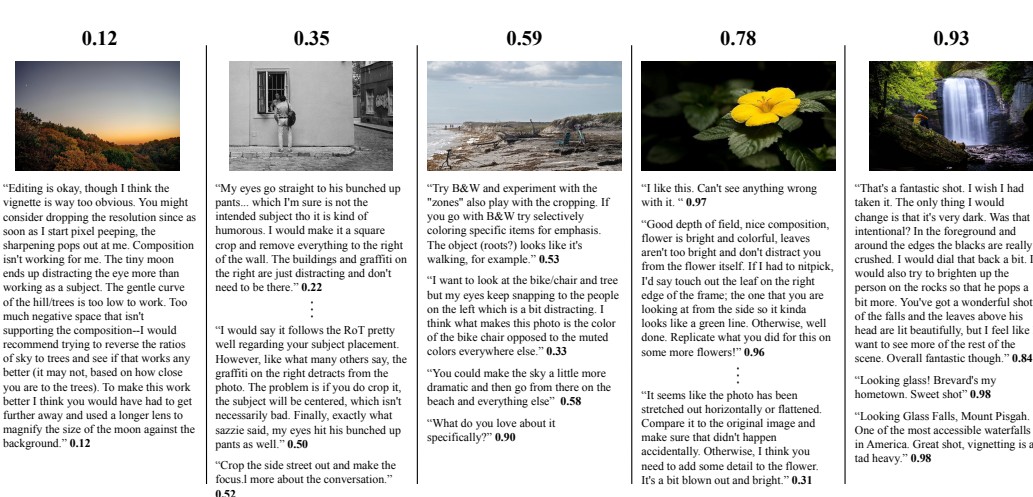

Figure 1: RPCD samples annotated with the proposed sentiment score. Sentiment scores are also reported for each comment.

of the bias introduced by the selected model, which was not present in the original work [1], would be beneficial. We exploit the implementation of TwitterRoBERTa finetuned for the sentiment prediction task vailable in the HuggingFace transformers library [39]. Figure 1 reports some samples of our dataset annotated with sentiment scores. Individual scores per critique are also included. It can be seen that most of the comments are focused on compositional and stylistic aspects of the photo. We estimate the sentiment score of the comments of AVA and PCCD for comparison. Figure 2 shows the distribution of sentiment scores for the samples of AVA, PCCD and our RPCD. We observe that the vast majority of the AVA and PCCD dataset samples are characterized by high sentiment scores, which produce left-skewed score distributions. On the other hand, the samples of the proposed RPCD cover almost the entire range of values with two peaks close to the values 1 and 0.5. This difference between ours and the other datasets indicates that RPCD have a richer representation of the whole aesthetic taste spectrum, providing information about why an image have a specific score for high and low sentiment scores. This dissimilarity can be explained by the nature of the different sources of each dataset. While DPChallenge is a community where users score each image, they are not encouraged to critique them as in the r/photocritique subreddit. Consequently, we hypothesize, this produces that only those users with a praise would leave a comment. The fact that there are many more users giving a score than commenting supports this possible explanation. The case of the PCCD

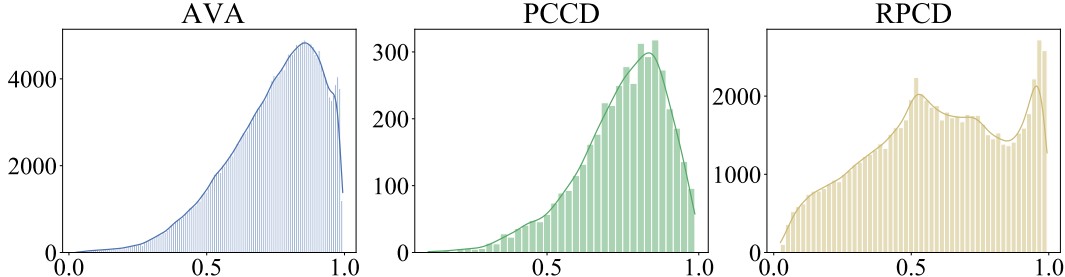

Figure 2: Sentiment score distribution on AVA, PCCD and our RPCD dataset.

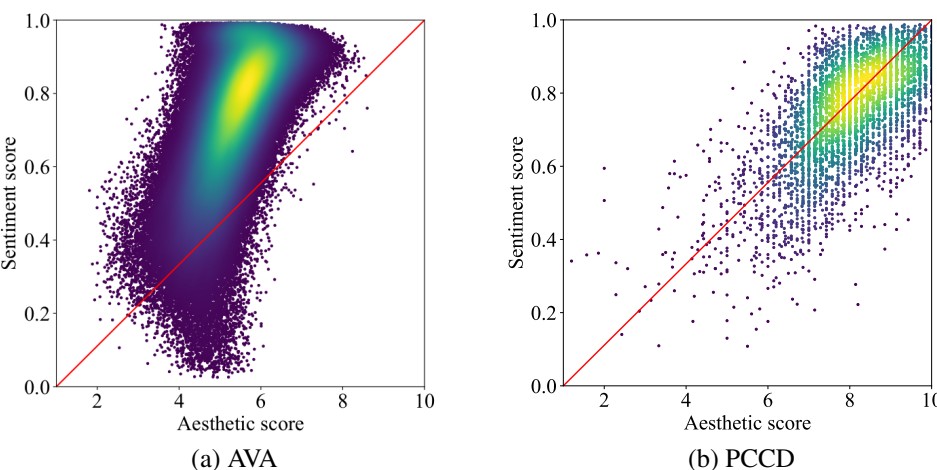

(a) AVA            (b) PCCD

Figure 3: Annotated aesthetic score *vs.* Sentiment polarity score for (a) AVA and (b) PCCD samples.

dataset is more difficult to analyze since the source website[8] does not provide the critiques feature anymore, but the fact that the dataset is heavily imbalanced may explain why the sentiment score is also imbalanced.

We also estimate how the sentiment score correlates with the annotated human aesthetic judgment for the AVA and PCCD images. In particular, we measure Spearman's Rank Correlation Coefficient (SRCC) and Pearson's Linear Correlation Coefficient (PLCC). On the AVA dataset, the SRCC is equal to 0.6418 while PLCC corresponds to 0.6424. For PCCD, SRCC is 0.6066 and PLCC 0.6499. The positive correlation on both datasets indicates the effectiveness of the proposed score and, therefore, that it represents a trustworthy approximation of the aesthetic score. Figure 3 shows the scatter plots relating the aesthetic score and sentiment score for the two considered datasets. It can be seen that most of the AVA aesthetic scores were originally annotated around the average value of the rating scale, i.e. 5. In fact, it is worth noting that AVA sentiment scores span the whole rating scale for aesthetic scores equal to or close to 5. We deepen the latter case in Appendix A.1. PCCD original scores, on the other hand, are very positive with a high concentration of samples for values between 7 and 9. Generally, our sentiment scores take on less biased values than previous aesthetic scores.

### 4.3 Content Analysis

In this section, we present an in-depth analysis of the images and comments in our dataset. This analysis is conducted by training different models to annotate aspects related to the semantics and composition of images and to estimate the usefulness and topics of the comments.

**Image Analysis.** For semantic content analysis, we group images into six categories, i.e., Animal, Architecture, Human, Landscape, Plant, and Static/Others. The semantic categories above are the

---

[8] https://gurushots.com/

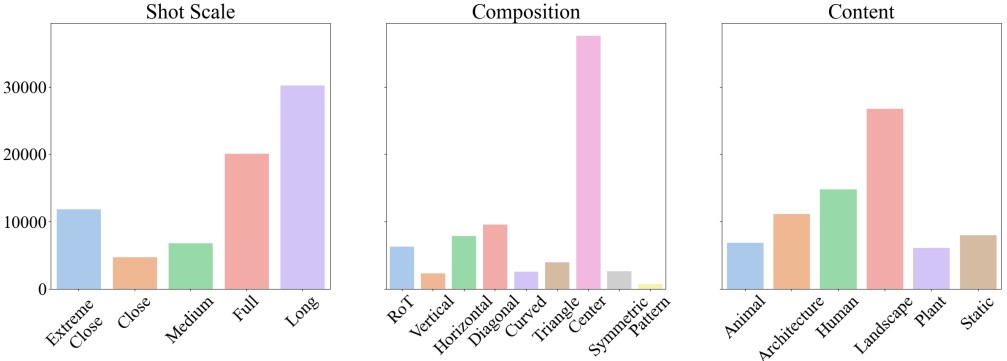

Figure 4: RPCD analysis. Left: Shot scale. Center: Image Composition, Right: Image Content.

same as CUHK-PQ, excluding the Night category. The latter is misleading as it represents the time of the shot and not a semantic category. Figure 4 shows the distribution of images per semantic category. As it is possible to see, most RPCD images contain landscapes. The second largest content category (approximately 15K images) includes human beings. The semantic class with the lowest number of instances is Plant. We also report results for shot scale classification, which determines the portion occupied by the main subject with respect to the frame. We distinguish five shot scale types defined in the training dataset MovieNet [15]: Extreme close-up, Close-up, Medium, Full, Long. Figure 4 shows that about 30K images have been labeled as Long shot scale. This result is in line with the fact that most images are of the semantic Landscape type. Very few images have been classified as Close-up or Medium range content, meaning photographers have preferred to capture subjects from very close or far away. Finally, we inspect images from the point of view of photographic composition. Specifically, images are categorized with respect to the main composition rule among the following eight: Rule-of-Thirds, Vertical, Horizontal, Diagonal, Curved, Triangle, Center, Symmetric, Pattern. The above composition rules are defined in the KU-PCP dataset [21]. Figure 4 presents the distribution of the images with respect to the composition rules. The images in our dataset are very bias on the Center category, indicating that in most images there is the main subject occupying the central region of the image. In Appendix A.4 we detail how we build the models for running the previous analysis and present some sample images for each of the analyzed aspects.

**Comment Analysis.** We analyze the topics of the corpus of comments in our dataset using BERTopic [12], a topic modeling technique. The most common topics in the analyzed datasets regards semantic aspects such as face, tree, bird, flower, and stars. There are also topics related to technical aspects of photography, namely ISO, aperture, dynamic range, and HDR. We refer to Appendix A.6 for a deeper analysis. Additionally, we use the definition of informativeness score of a previous work [10] to estimate whether the comments of our dataset are meaningful and how do they compare to other datasets. In the Appendix A.7 we compare the results on our dataset with those of state-of-the-art datasets, finding that, on average, the proposed RPCD contains the most informative comments, with an informativeness score slightly higher than PCCD and more than double than AVA.

## 5 Evaluation

To illustrate the possible uses of the newly introduced dataset, we run several experiments around the image aesthetic assessment task, where our goal is to predict an aesthetic score given an image, as well as in the image captioning task, where our goal is to predict an aesthetic critique given an image. To this aim, we split the whole dataset into 70% training samples, 10% validation samples, and the remaining 20% for testing.

## 5.1 Image Aesthetic Assessment

The main motivation to create this dataset is to perform Image Aesthetic Assessment with the interpretability of the aesthetic critiques. We use the scores computed using sentiment analysis and propose a method to predict such scores. We also run SOTA models on our dataset for comparison.

In Table 2 we report SRCC, LCC and Accuracy on our RPCD, PCCD [5] and AVA [28] datasets using the sentiment score, comparing the results of `NIMA` [37] and other experiments we carried out. We additionally perform an extensive evaluation of the family of ViT models in Appendix B to assess the suitability of such models for the aesthetic assessment task, evaluating its performance on AVA dataset. We highlight that ViT Large (we called `AestheticViT`) outperforms previous SOTA model [14] by a 4% in the correlation metrics on AVA dataset using the original scores. In the `ViT + Linear probe` experiments, we also study to what extent the results obtained to predict the aesthetic and sentiment scores are due to the knowledge already present in the pretrained model. The accuracy is computed defining as high quality images those with an score above 5, and poor quality otherwise. The results in Table 2 show that, although `ViT + Linear probe` and `AestheticViT` outperform a previous aesthetic model used as a baseline, it does not achieve a good enough performance in any of the benchmarks to predict the proposed sentiment score. Moreover, the training of the model on AVA deteriorate its performance. This results proofs how challenging the task is and may indicate that the main previous benchmark, AVA, may be biased towards the content of the images, reducing the importance of the actual aesthetics of the photo. We would like to support the reasoning of Hosu *et al*. [14] regarding the suitability of correlation metrics rather than accuracy to evaluate this task. While correlation metrics are representative of the entire range of scores, image labels ('good' or 'bad') are defined arbitrarily, which becomes an issue when the label distribution is imbalanced as in the case of AVA and PCCD datasets for both the original and sentiment scores.

Table 2: Sentiment score baseline on the three considered datasets.

| Method | AVA | | | PCCD | | | RPCD | | |
|---|---|---|---|---|---|---|---|---|---|
| | SRCC | LCC | Acc. (%) | SRCC | LCC | Acc. (%) | SRCC | LCC | Acc. (%) |
| NIMA [37] | 0.253 | 0.259 | 90.20 | 0.066 | 0.070 | **93.87** | 0.120 | 0.116 | 63.25 |
| ViT + Linear probe* | **0.570** | **0.570** | 76.43 | 0.156 | 0.165 | 93.04 | 0.172 | 0.173 | 64.58 |
| AestheticViT* | 0.544 | 0.550 | **90.54** | **0.228** | **0.262** | 93.86 | **0.250** | **0.253** | **65.27** |

*Best performing models. See Appendix B

## 5.2 Aesthetic Critiques

We also evaluate our dataset on the task of Aesthetic Image Captioning (AIC), using a SOTA model [22]. To evaluate the results, we follow the procedure of [5] as our dataset also contains more than one reference caption that corresponds to a single image. Table 3 compares the obtained results with the previous work [5] we use as reference. We observe that the achievable performance is far lower than that obtained for the description of the image content (See COCO captions benchmark [9]), and further work is necessary to produce meaningful aesthetic captions. More details on the aesthetic critique procedure can be found in Appendix B.2.

Table 3: Aesthetic image captioning using BLIP [22] on PCCD and our RPCD.

| | Bleu1 | Bleu2 | Bleu3 | Bleu4 | METEOR | ROUGE | CIDEr | SPICE |
|---|---|---|---|---|---|---|---|---|
| PCCD | 0.165 | 0.065 | 0.028 | 0.011 | 0.063 | 0.137 | 0.049 | 0.048 |
| RPCD | 0.211 | 0.088 | 0.038 | 0.017 | 0.077 | 0.157 | 0.048 | 0.040 |

## 6 Discussion and Future Work

We presented the Reddit Photo Critique Dataset (RPCD) consisting of image and photo critiques tuples. This dataset is collected by crawling posts from a community where people are encouraged to

---

[9]https://paperswithcode.com/sota/image-captioning-on-coco-captions

criticize positive and negative image aesthetic aspects. Our dataset has approximately 18× the images and 7× the comments compared to the PCCD dataset. Compared to AVA, the best-known aesthetic assessment dataset, our RPCD has longer and more meaningful comments and higher resolution images. Together with the dataset, we have for the first time in the literature defined an approach to obtain the aesthetic ranking of images directly from the analysis of comments. The proposed approach is based on the sentiment analysis of the comments. The proposed score was shown to have a positive correlation with the aesthetic judgments of humans. Therefore, RPCD can be used both to predict aesthetic captions and to estimate an aesthetic score. We conducted several experiments for the image aesthetic assessment task in which we compared the results obtained from different methods on our dataset, AVA and PCCD. These experiments show that a ViT is able to obtain good performance on AVA while both PCCD and RPCD are more challenging. The use of content-aware (ViT + Linear Probe), instead of aesthetic-aware (Aesthetic ViT), features results in a significant drop in performance for those datasets that may be less content-biased. Experiments on aesthetic image captioning carried out on PCCD and RPCD datasets highlight that the achievable performance is far lower than that obtained for the description of the image content, and further work is necessary to produce meaningful aesthetic captions.

However, several limitations remain. First, the limited number of comments per image (i.e., 3 in average), although the comments are long and very informative, could make the evaluation biased by the few users and not sufficiently objective. Second, we encourage the Machine Learning community to work on alternative or complementary solutions to the proposed sentiment analysis as a proxy for aesthetics. Among other things, the fact that the sentiment score is automatically estimated could cause noisy annotations and the change in the data domain should be further studied. This noisy annotations could be influenced by the potential bias present in the selected model. Third, the aesthetic captioning task remains an open challenge. Finally, the concept of aesthetics expressed in our dataset must be understood limitedly to the Western cultural and geographical context on the basis of the demographic statistics of the Reddit users (see Appendix A.1). Additionally, other ethical considerations are discussed in Appendix D.

Despite the above limitations, we believe RPCD is an important contribution for the design of multi-modal and explainable aesthetic assessment models. As a future work, we would like to deepen the ranking method based on the analysis of comments in order to make it more reliable and diversified for the different aspects that characterize the aesthetics: style, color, composition, etc. In addition we will be able to interpret which are the aspects that are evaluated more positively or negatively by users. Finally, exploiting new sources of available data may provide further benefits while training larger models.

## Acknowledgments

This project is supported by Ringier, TX Group, NZZ, SRG, VSM, Viscom, and the ETH Zurich Foundation.

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
