# OpenReview forum: "Understanding Aesthetics with Language: A Photo Critique Dataset for Aesthetic Assessment"
_NeurIPS.cc/2022/Track/Datasets_and_Benchmarks — NeurIPS 2022 Datasets and Benchmarks _

### Official Review · Reviewer_KEe2 · 2022-07-06
**A new photo critique dataset**

**Rating:** 5
**Confidence:** 4
**Correctness:** seems ok.

**Strengths:**

1. A new photo critique dataset that is considerably larger and richer ( in terms of comments) than existing ones

2. A simple ranking algorithm for image aesthetic assessment

3. Reasonable experimental validation and comparison with other datasets.

**Weaknesses:**

1. There is not sufficient details pertaining to dataset documentation - as outlined in datasheets/artsheets paper. Please also refer to comments in the sections below.  Please see detailed comments under ethics and documentation sections.

2. Presentation clarity can be improved- fixing typos, elaborating certain sections, etc. - please see comments under clarity.

**Additional Feedback:**

Please see responses to above questions.

**Clarity:**

- There is reference to Appendix A and B, but these are not included with the submission.

- some typos can be found in the paper- e.g., In Section 4.1, "containsts"  should be contains

- Certain sections could be elaborated to provide more clarity and details. For e.g., sec 3.3 can be elaborated to emphasize how the proposed method is better than existing approaches - how the method allows weak supervision, which two aesthetic tasks does it combine and why that is useful,  and on what grounds it is a strong indicator of human judgement.

**Documentation:**

The paper provides some information regarding the data collection and organization, that said, it does not contain a detailed data sheet that outlines the data composition, provenance, maintenance, use, distribution, and the like. Information related to data consent is also not provided. Although the subreddit included in the dataset comes from photographers and amateurs, there is not enough information related to these annotators --as mandated in typical dataset documentations.  The paper might benefit by providing a detailed response to all questions in the data sheets for dataset paper, and more specifically from art sheets for art datasets paper as the proposed dataset concerns photography- which can also be a form of art.

**Ethics:**

Ranking based on "image aesthetics " as noted in the paper is a subjective task and can raise feasibility questions such as judging skills of photographers algorithmically basing on metrics which may or may not be reliable. Although this may not be the intended application of this paper, the dataset could be leveraged for such a task. Thus, it becomes all the more important to discuss in detail questions related to use and distributions and all other questions as outlined in datasheets and art sheets papers.

**Relation To Prior Work:**

The paper needs to be contextualized with respect to prior dataset documentation papers such as datasheets for datasets and art sheets for art datasets.

**Summary And Contributions:**

The paper introduces a dataset for the purposes of image aesthetic assessment. Specifically, the dataset called Reditt photo Critique Dataset (RPID) consists of 74k images with accompanying 220 k (user) comments /critiques which serve as a richer source of aesthetic characteristics than mere ratings which can be subjective and biased.  The main distinction of the proposed photo critique dataset when compared to existing photo critique dataset is that the images are mostly Full HD. Apart from this, it is bigger and richer in terms of comments and number of images.  The paper also proposes an image ranking algorithm based on the aesthetic assessment data available.

---

> ### Author Response · Authors · 2022-08-23
> **Our response to Reviewer KEe2**
>
> We thank the reviewer for the valuable feedback, which has allowed us to improve our paper. We have carefully looked for typos and corrected them. Below we treat his comments as exhaustively as possible, while all changes in the article are highlighted in blue.
>
> > __*C1: There is reference to Appendix A and B, but these are not included with the submission.*__
>
> Please note that all appendixes are included in the Supplemental Material of our submission for the NeurIPS Datasets and Benchmark track on OpenReview. Furthermore, same versions of the submitted main article and supplementary material are also accessible in the following ArXiv pre-print: https://arxiv.org/abs/2206.08614.
>
> > __*C2: Certain sections could be elaborated to provide more clarity and details. For e.g., sec 3.3 can be elaborated to emphasize how the proposed method is better than existing approaches - how the method allows weak supervision, which two aesthetic tasks does it combine and why that is useful, and on what grounds it is a strong indicator of human judgment.*__
>
> Due to the limit of pages set at 9, we had chosen to reserve more space for the presentation of the dataset. Now that we have been granted the addition of the tenth page we have expanded Section 3.3 to deepen the proposed method.
>
> > __*C3: Documentation & Data consent*__
>
> As pointed out by the reviewer, this paper might be improved by adding a detailed response to the questions proposed in Datasheets for Datasets [1]. Hence, to address the concerns regarding the dataset documentation, we have added the datasheet detailing our proposed dataset in Appendix F, which we have also included in the Github repository.
> Regarding the data consent, and more broadly, the license of the data, we have modified the way to access the data. Instead of providing the files including the text and other metadata from each of the posts and comments, we only provide the ids of the posts used to build the dataset along with the code necessary to retrieve it directly from Reddit. This way we make sure that the Reddit user has complete control over the data, and we limit our work to retrieve the publicly available data using the appropriate tools (Reddit API and Pushshift API). The downside of this is that the dataset will not be exactly the same as time goes by, but it is unlikely to suffer substantial changes (removing posts, comments, etc).
>
> [1] Timnit Gebru, Jamie Morgenstern, Briana Vecchione, Jennifer Wortman Vaughan, Hanna Wallach, Hal Daumé Iii, and Kate Crawford. Datasheets for datasets. Communications of the ACM, 64(12):86–92, 2021.
>
> > *__C4: Ethics__*
>
> We hope to have appropriately addressed the ethical concerns in Appendix D and Appendix F. Additionally, for the sake of expanding the number of potential ethical concerns considered, we have added the case commented by the reviewer to the Appendix F.5 about dataset uses.

---

### Official Review · Reviewer_uNZv · 2022-07-23
**A very good dataset paper**

**Rating:** 6
**Confidence:** 3
**Correctness:** The submissison looks to me technical…
**Clarity:** Yes

**Strengths:**

1. The work looks technically sound.
2. Technical details of data collection, implementation and evlauation are provided.
3. The paper is well-written and easy to follow

**Weaknesses:**

1. Contributions of the work can be further elaborated
2. Ethics issues have been discussed but some issues are not clear to me.
3. Data access can be arranged in a more user friendly way.

**Additional Feedback:**

NA

**Documentation:**

Yes

**Ethics:**

Ethics issues have been discussed in the supplementary material. However, some issues are still not clear to me. For example, data sourced from third parties has to comply with copyright law and/or terms and conditions present at the original source. Do the authors have permission to share these images under an open licence? Who is the copyright owner? How robust are the methods used to comply with GDPR? Can the authors guarantee that no personal or identifying data has been shared in this dataset?


**Relation To Prior Work:**

Yes. Table 1 also provides a good comparison of the proposed dataset with the benchmarked datasets.

**Summary And Contributions:**

This paper presents the Reddit Photo Critique Dataset (RPCD) consisting of image and photo critiques
tuples. The dataset was collected by crawling posts from a community where people posted positive and negative comments on image aesthetics. Detailed data collection techniques/processes and the aesthetics evaluation are provided. Overall, the paper is well-written and easy to follow. Aesthetics is always an important topic in design studies and computer vision studies. Therefore, the dataset may have great application potential.

---

> ### Author Response · Authors · 2022-08-23
> **Our response to Reviewer uNZv**
>
> Thanks for the review. We appreciate the comments and the raised concerns. Below we treat his comments as exhaustively as possible, while all changes in the article are highlighted in blue.
>
> >__*C1: Contributions of the work can be further elaborated*__
>
> We have expanded and highlighted the key contributions of our work in the Introduction section.
>
> >__*C2: Ethics*__
>
> We understand the data consent concerns and, more generally, the GDPR compliance of the dataset. For this reason, and as detailed in the comment to Reviewer KEe2, we have modified the way to access the data. Instead of providing the files including the text and other metadata from each of the posts and comments, now we only provide the ids of the posts used to build the dataset along with the code necessary to retrieve it directly from Reddit. This way we make sure that the Reddit user has complete control over the data, and we limit our work to retrieve the publicly available data using the appropriate tools (Reddit API and Pushshift). The downside of this is that the dataset will not be exactly the same as time goes by, but it is unlikely to suffer substantial changes.
>
> We also want to address each of the points raised in the review:
> * __*Do the authors have permission to share these images under an open license?*__ We have not shared in any moment the images, only the url to the images that are hosted in a third party service (such as Imgur or Reddit itself).
> * __*Who is the copyright owner?*__ According to the [Reddit API Terms of Use](https://www.reddit.com/wiki/api-terms/#wiki_api_terms_of_use), is the user: "User Content. Reddit user photos, text and videos ("User Content") are owned by the users and not by Reddit.”
> * __*How robust are the methods used to comply with GDPR?*__ With the new way to access the data, we do not provide any kind of personal information, and we rely on Reddit and Pushshift to retrieve the data. Thus, the user may use the GDPR compliance methods provided by these services.
> * __*Can the authors guarantee that no personal or identifying data has been shared in this dataset?*__ Now that we only share the post ids, no personal information is contained in the provided dataset. However, we provide the code to retrieve publicly available data from Reddit that may contain personal or identifiable information. This data is under the control of the Reddit user that may decide to remove it from Reddit and, thus, remove it from the dataset.

---

### Official Review · Reviewer_8ptL · 2022-07-27
**Paper introduces a high-impact dataset that addresses existing issues in aesthetic analysis**

**Rating:** 7
**Confidence:** 3
**Correctness:** Yes.
**Clarity:** Yes.

**Strengths:**

- The provided dataset has the potential to be quite useful to the computer vision and NLP communities. Including both scalar values as well as full comments allows for a wide range of future work to be pursued.
- The dataset appears to be thoughtfully constructed and the development process is well documented in the text.
- The paper is very well written: It establishes sufficient motivation, carefully outlines the thought process of the authors, and answers many potential questions a reader may have. The text includes a nice analysis of limitations and future work.
- The figures, graphs, and examples are well chosen.

**Weaknesses:**

- It would be interesting to compare the authors’ approach to assigning a scalar value measuring the aesthetics of an image to the scores a human would choose to assign to the image when prompted, given that the latter seems to be the primary method for labeling among most aesthetics datasets.
- As noted by the authors, given the high subjectivity of an aesthetic evaluation of a photograph it is unlikely that the average of 3 comments per image fully captures the distribution of human sentiment for a given data point.


**Additional Feedback:**

N/A

**Documentation:**

Yes.

**Relation To Prior Work:**

Yes, although it would be potentially helpful to include prior work from a broader range of research areas. For example, citing relevant work in sentiment prediction outside image aesthetic assessment.

**Summary And Contributions:**

The paper introduces the RPCD which consists of 74k high-quality photos paired with critique comments from the r/photocritique subreddit. The authors introduce an algorithm for extracting quantitative aesthetic ratings from these comments and evaluate models’ ability to predict these scores

---

> ### Author Response · Authors · 2022-08-23
> **Our response to Reviewer 8ptL**
>
> Thank you for your time. We highly appreciate the comments of the reviewer and we hope to address the key comments below. All changes in the article are highlighted in blue.
>
> > __*C1: It would be interesting to compare the authors’ approach to assigning a scalar value measuring the aesthetics of an image to the scores a human would choose to assign to the image when prompted*__
>
> One of the main points of the proposed approach was to find ways to use non-annotated data that can be available on the Internet. For that reason, we have not considered manually annotating the proposed dataset with aesthetic scores. Instead, we use the comments to build an annotation. However, we compute the automated sentiment-based score on the two relevant datasets, AVA and PCCD, and we compare them to the human annotations. We discuss more about this in Section 4.2 (last paragraph) and show the relation between the automatic and human annotated scores in Figure 3.
>
> >__*C2: As noted by the authors, given the high subjectivity of an aesthetic evaluation of a photograph it is unlikely that the average of 3 comments per image fully captures the distribution of human sentiment for a given data point.*__
>
> This is a very valid point we agree with, as noted in the discussion about limitations of the work. However, we would like to highlight the potential to build explanatory systems that do not reduce the aesthetic assessment to a score (although we show that it is also possible).
>
> >__*C3: Citing relevant work in sentiment prediction outside image aesthetic assessment.*__
>
> We modified Section 4.2 by adding a brief overview of the relevant work in sentiment analysis. Furthermore, we have better motivated the choice of the model used to obtain the predictions on the aesthetic critiques of our dataset.

---

### Official Review · Reviewer_2Q2e · 2022-07-28
**A sound contribution to Automated Image Aesthetics Assessment**

**Rating:** 7
**Confidence:** 2
**Clarity:** The paper is written in a clear way.

**Strengths:**

Compared to already existing datasets, RPCD offers some advantages, including 1) higher image resolution and 2) higher average word per comment (albeit only slightly higher than PCCD). The paper (including the appendices) also presents the methodology and the results of the experiments in a clear way.


**Weaknesses:**

The authors already identified some of the limitations of the paper (p. 9), such as the limited number of comments per image and the risks associated with the automated labelling of aesthetic judgment in the dataset. Another weakness of this paper is that the use of the sentiment polarity of the comments as a proxy for the image aesthetic score is not justified in a clear way. In particular, compared to the other datasets presented in Table 1, RPCD appears to be limited by the use of a rating scale on a 0-1 range (based on the sentiment polarity) rather than a 0-10 scale. Some analysis of the comparison of the different scales would have been useful.

**Additional Feedback:**

No additional feedback.

**Correctness:**

The dataset is constructed in a sound way and the experiments are designed and reported in an appropriate way. However, compared to other benchmark dataset for aesthetic image captioning, this dataset is characterised by the fact that the aesthetic score was automatically assigned (machine-based annotation). A more detailed discussion of the limitations of this approach (which are acknowledged by the authors) would have been useful

**Documentation:**

The documentation is sufficient.

**Ethics:**

The authors state they complied with Reddit User Agreement, Reddit Privacy Policy, Reddit API terms of use and PushShift database Creative Commons License. However, the authors did not conduct any analysis about the possible presence of bias or offensive content in the dataset , as they deemed outside the scope of their work. However, at least a limited analysis of these issues would have been beneficial.

**Relation To Prior Work:**

The paper clearly discusses how this work relates to previous contributions and it includes experiments comparing RPCD with other banchmark datasets.

**Summary And Contributions:**

This paper introduces the Reddit Photo Critique Dataset  (RPCD), a new dataset of 74K high-resolution images and 220 associated comments from photography groups on Reddit. An aesthetic score / judgment is assigned to each image based on the sentiment polarity score of the critique of the image contained in the comments. The sentiment polarity score is assigned automatically calculated using TwitterRoberta. The paper also includes the results of a series of experiments that the authors conducted to illustrate the potential uses of the dataset.

---

> ### Author Response · Authors · 2022-08-23
> **Our response to Reviewer 2Q2e**
>
> We are grateful for the feedback provided by the reviewer, and we would like to address the comments below. All changes in the article are highlighted in blue.
>
> >__*C1: Another weakness of this paper is that the use of the sentiment polarity of the comments as a proxy for the image aesthetic score is not justified in a clear way. In particular, compared to the other datasets presented in Table 1, RPCD appears to be limited by the use of a rating scale on a 0-1 range (based on the sentiment polarity) rather than a 0-10 scale. Some analysis of the comparison of the different scales would have been useful.*__
>
> We have expanded Section 3.3 to better justify the design of the computed aesthetic score. We do not consider the rating scale a problem, but simply a characteristic of our dataset. Moreover, in a comprehensive survey on computational aesthetic Evaluation [1], it is possible to see how in the literature the ranges chosen are peculiar to each dataset. For example, the 0-10 range of AVA comes from the scale used on the platform when acquiring user ratings. Our score was defined in the way we considered most reasonable. However, it is still a first attempt for mapping photo critiques into an aesthetic score as specified in the manuscript. In light of the previous considerations, we believe that a multiplication factor of 10 (which could be 100 or other arbitrary values) is a difficult choice to justify and would not affect the learning process.
>
> [1] Jiajing Zhang, Yongwei Miao, and Jinhui Yu. A comprehensive survey on computational aesthetic evaluation of visual art images: Metrics and challenges. IEEE Access, 2021.
>
> >__*C2: A more detailed discussion of the limitations of this approach (which are acknowledged by the authors) would have been useful.*__
>
> We have expanded the discussions of the limitations of the approach (blue text) in Section 6. In particular, we added the consideration that the concept of aesthetics in our dataset must be understood exclusively with regard to the geographical context of the majority of Reddit users, namely the Western people. The latter aspect was ascertained following the analysis of the statistics on Reddit members reported in the Appendix A.1. Any additional points we should cover or any further comments on this are appreciated.
>
> >__*C3: the authors did not conduct any analysis about the possible presence of bias or offensive content in the dataset , as they deemed outside the scope of their work. However, at least a limited analysis of these issues would have been beneficial.*__
>
> We include in the Appendix D “Ethical considerations” a short analysis using a state of the art library to identify offensive content, Detoxify [1], which is detailed in Appendix A.5 Explicit or offensive content. This analysis is far from comprehensive, but it is a first step to understanding the kind of content present in the dataset and opens the possibility to easily filter out those comments considered offensive.
>
> [1] Laura Hanu and Unitary team. Detoxify. Github. https://github.com/unitaryai/detoxify, 2020.

---

### Review · Ethics_Reviewer_NqU4 · 2022-08-21

**Recommendation:** 2

**Ethics Documentation:**

Appendix D is helpful but should be referenced in the paper itself.

Cultural relativity: Aesthetic notoriously subjective and often influenced by cultural beauty standards such as eurocentrism in facial beauty. The introduction appropriately acknowledges the lack of explanation for aesthetic scores

Bias in dataset/Explicit/offensive content: a preliminary bias analysis is necessary to understand the limitations of how their findings can be used.

Bias in labelers/Make-up of labelers: Authors should have some explanation on the backgrounds and makeup of the communities writing these descriptions and biases that come with a specific online community like Reddit.

Biases in tools: some explanation on why the authors choose TwitterRoberta and not other sentiment analysis models would be helpful. While this tool is trained for social media text, Twitter is a different format and community from Reddit and also sentiment analysis models have their own sets of biases (which should be addressed in the paper or appendices).

Privacy: While the onus is ultimately on Reddit to ensure privacy and content moderation of the users and content, a public dataset does not inherently mean that the subjects consented to their data being used, especially for this task. The consent piece should be further acknowledged in Appendix D.

**Ethics Review:**

- Cultural relativity
- Bias in dataset
- Bias in tools
- Bias in labelers
- Make-up of labelers
- Privacy
- Explicit/offensive content

---

> ### Author Response · Authors · 2022-08-23
> **Our response to Ethics Reviewer NqU4**
>
> We would like to thank the reviewer for his important comments regarding ethical issues. We have promptly responded to the issues raised below and highlighted the changes made to the manuscript in blue.
>
> > *__Appendix D is helpful but should be referenced in the paper itself.__*
>
> Appendix D is now referenced in the discussion section (Section 6).
>
> > *__Bias in dataset/Explicit/offensive content: a preliminary bias analysis is necessary to understand the limitations of how their findings can be used.__*
>
> We have included a preliminary analysis of the presence of offensive content in the dataset in Appendix A.5, which is referenced in the Ethical Considerations appendix. This analysis is far from comprehensive, but it is a first step to understanding the kind of content present in the dataset and opens the possibility to easily filter out those comments considered offensive.
>
> > __*Bias in labelers/Make-up of labelers: Authors should have some explanation on the backgrounds and makeup of the communities writing these descriptions and biases that come with a specific online community like Reddit.*__
>
> We gathered some information about the gender, age group, and country of Reddit members. We found that there is a prevalence of men of Western origin and aged between 18 and 29 years. In light of the previous statistics, it is necessary to underline that the data treated in our dataset, therefore the inferred concept of aesthetics, presents a bias due to the limited cultural and geographical integration of the people who produced the information. We added the previous considerations in the revised manuscript by reporting the statistics and aforementioned consideration in Appendix A.1. Furthermore, we mentioned the presence of the member bias among the limitations of our dataset presented in Section 6.
>
> >*__Cultural relativity: Aesthetic notoriously subjective and often influenced by cultural beauty standards such as eurocentrism in facial beauty. The introduction appropriately acknowledges the lack of explanation for aesthetic scores__*
>
> We totally agree with the reviewer that given the extreme subjectivity of aesthetics it is necessary to deepen the aspect concerning the geographical, social and cultural background of the members of Reddit. Therefore, we have tried to clarify this aspect in the light of the statistics collected to answer the previous comment. The concept of aesthetics expressed in our dataset should be limited to that of Western people. In the revised manuscript we highlighted this issue in the limitations of the dataset (Section 6) and Appendix A.1.
>
> > *__Biases in tools: some explanation on why the authors choose TwitterRoberta and not other sentiment analysis models would be helpful. While this tool is trained for social media text, Twitter is a different format and community from Reddit and also sentiment analysis models have their own sets of biases (which should be addressed in the paper or appendices).__*
>
> In Section 4.2 we have added a brief literature on sentiment analysis methods. Next we introduce the TwitterRoBERTa method, motivating the choice. Finally we acknowledge the limitations of the selected method, which are also mentioned in the discussion section.
>
> > __*Privacy: While the onus is ultimately on Reddit to ensure privacy and content moderation of the users and content, a public dataset does not inherently mean that the subjects consented to their data being used, especially for this task. The consent piece should be further acknowledged in Appendix D.*__
>
> Data consent concerns have been addressed in Appendix D with an additional point on this, and, additionally, in the respective questions of the datasheet available in Appendix F.

---

### Meta-Review · Area_Chair_agCq · 2022-09-06

**Recommendation:** Accept
**Confidence:** 5

**Metareview:**

This paper introduces a new dataset for critiquing photos mining ~74K from the r/photocritique subreddit and an algorithm for transforming the reddit comments into a judgement score between 0-1. They then introduce the task of predicting these judgement scores from the images. All reviewers find this to be a useful contribution to the community, and they appreciate the thoughtful process behind the dataset creation process as well as the thorough documentation of this process in the paper. All reviewers also find the manuscript well written and easy to understand. The authors have also addressed all points brought up by the ethics review.

---

### Decision · Program_Chairs · 2022-09-16

Accept